# Rapid Joining of Commercial-Purity Ti to 304 Stainless Steel Using Joule Heating Diffusion Bonding: Interfacial Microstructure and Strength of the Dissimilar Joint

**K. T. Suzuki, Y. S. Sato *** and S. Tokita

Department of Materials Processing, Graduate School of Engineering, Tohoku University, Sendai 980-8579, Japan; kiyoaki.suzuki.q1@dc.tohoku.ac.jp (K.T.S.); shun.tokita.c4@tohoku.ac.jp (S.T.)
* Correspondence: ytksato@material.tohoku.ac.jp; Tel.: +81-22-795-7352

**Abstract:** A new solid-state bonding technique, Joule heating diffusion bonding, was used for the dissimilar bonding of commercial-purity Ti to 304 stainless steel within a short time without macroscopic deformation of the workpieces. The tensile strengths of the joints produced at various bonding parameters were examined at room temperature, and the microstructures of the joints and the fracture surfaces were analyzed to clarify the effect of the microstructural factors on the tensile strength of the joints. The tensile strength of the joints increased with the increase in the fraction of the sufficiently bonded interface. In the joints with the well-bonded interface, the tensile strength decreased with the increase in the thickness of the brittle Fe-Ti-type intermetallic compound layers at the joint interface. This study suggested that the high tensile strength could be achieved in the Joule heating diffusion bonded joints with the well-bonded interface where the thickness of the Fe-Ti-type intermetallic compound layers was thinner than 0.5 μm.

**Keywords:** Joule heating diffusion bonding; dissimilar bonding; commercial-purity titanium (CP-Ti); 304 stainless steel; mechanical property; microstructure

---

## 1. Introduction

Dissimilar joining of titanium (Ti) to stainless steel is an effective method to fabricate the cost-effective and high-performance parts and structures for the offshore industry [1]. However, the conventional fusion welding techniques, such as arc welding, cannot be utilized for the dissimilar bonding of Ti to stainless steel [2]. These methods result in the fabrication of joints with poor mechanical properties owing to the formation of diverse brittle intermetallic compounds (IMCs) at the interface [3]. Satoh et al. demonstrated that the joints that were fabricated by the laser welding of commercial-purity titanium (CP-Ti) and austenitic 316 stainless steel exhibited low tensile strengths owing to the formation of intermetallic phases occupying the weld metal entirely [4]. The conventional fusion welding techniques also involve the melting and solidification of the base metals. This induces not only high residual stresses, owing to the difference in the coefficients of thermal expansion [3], but also crack formation during cooling. Furthermore, the joining of Ti is hindered by the easy dissolution of oxygen and nitrogen from the welding atmosphere during melting [5].

There has been extensive research on the dissimilar bonding of Ti to stainless steel using solid-state bonding techniques that prevent the formation and growth of IMCs [6]. Ghosh et al. studied the diffusion bonding of CP-Ti to 304 stainless steel at 850 to 950 °C for 2 h under a uniaxial pressure of 3 MPa. They detected the presence of numerous brittle IMCs, such as the σ phase (Fe-Cr-type compound), $Fe_2Ti$, $Cr_2Ti$, the χ phase (Fe-Ti-Cr ternary compound), FeTi, and $Fe_2Ti_4O$, and retained

β-Ti at the joint interface [7]. They also revealed that the joint at 850 °C exhibited the highest tensile strength; furthermore, the tensile strength decreased with the increase in the bonding temperature owing to the increase in the thickness of the brittle IMC layer that primarily comprised FeTi and Fe$_2$Ti [8]. Akbar et al. bonded CP-Ti to 316 stainless steel by diffusion bonding using copper foil as an interlayer at various bonding conditions. The maximum tensile shear strength was achieved at a bonding temperature of 950 °C and a holding time of 50 min [9]. Shirzadi et al. examined the gallium-assisted diffusion bonding for the dissimilar bonding of CP-Ti to 304L stainless steel. The abutted surfaces of the base materials were ground with emery paper containing a small amount of liquid gallium to remove the oxide layer before bonding and achieved a higher tensile strength of joints than the conventional diffusion bonding techniques [10]. Ananthakumar et al. studied the plasma-assisted diffusion bonding of CP-Ti to 304L stainless steel, and they successfully produced the joints for a relatively short time of 15 min at 650 °C [11]. Szwed et al. bonded CP-Ti and 304 stainless steel by transient liquid phase (TLP) bonding with an Ni interlayer at 950–1000 °C for 1 h under a compressive pressure of 2 MPa. They reported that it was very difficult to prevent the formation of Fe-Ti-type IMCs at the interface [12]. Akbarimousavi et al. studied the friction welding of CP-Ti/316L stainless steel and obtained the highest tensile strength at a friction pressure of 98 MPa for 4 s and a forged pressure of 331 MPa for 3 s [13].

The previous studies indicated the suitability of solid-state bonding techniques over fusion welding techniques for the dissimilar bonding of Ti to stainless steel. This is because solid-state bonding results in the elimination or decline in the problems that are associated with melting during fusion welding. However, the practical applications of solid-state bonding techniques are limited because it is difficult to induce bonding within a short time without the severe deformation of the workpiece. Friction welding, friction stir welding, and ultrasonic welding are achieved within a short processing time and inevitably result in the severe deformation of the workpieces owing to the relative motion of either the workpieces or the additional tool/part. Furthermore, the macroscopic deformation-free processes, such as diffusion bonding, require the long-term exposure of the workpieces to high temperatures. Therefore, the conventional solid-state bonding techniques hardly achieve both the macroscopic deformation of the workpieces and the processing time simultaneously.

A novel, effective solid-state bonding technique, Joule heating diffusion bonding (JHDB), was utilized for the dissimilar bonding of CP-Ti to 304 stainless steel in the present study. A newly developed JHDB system with high power source induces the local heating of the abutted surfaces to the target temperature, without overshooting, at a heating rate higher than 1000 °C/s. Therefore, JHDB results in the rapid fabrication of joints without the macroscopic deformation of the workpieces, which is a unique superiority of JHDB, which the other bonding processes using Joule heating could hardly achieve [14]. This results in the effective suppression of the formation and growth of IMCs during the dissimilar bonding of CP-Ti to 304 stainless steel. However, JHDB is a new bonding technique that has not been subjected to systematic fundamental analysis. The present study demonstrated the application of JHDB for the dissimilar bonding of CP-Ti to 304 stainless steel for various sets of bonding parameters and examined the microstructure and the tensile strength of the joints. The objective of the present study is to systematically elucidate the effect of the microstructural factors at the interface on the tensile strength of the joints produced by JHDB.

## 2. Experimental Procedures

### 2.1. Materials

Cylindrical rods (diameter = 8 mm) of CP-Ti (Grade 2) and commercial 304 stainless steel (X5CrNi18-10 in ISO 15510:2014) were utilized as the base materials in this study. The lengths of the CP-Ti and 304 stainless-steel samples were 30 mm and 50 mm, respectively. Tables 1 and 2 list the chemical composition of CP-Ti and 304 stainless steel used in the present study, respectively.

The abutted surfaces of both the base metals were prepared by machining and cleaned with ethanol before JHDB.

**Table 1.** Chemical composition (wt. %) of commercial-purity titanium (CP-Ti).

| Ti | Fe | N | O | H |
|---|---|---|---|---|
| Bal. | 0.04 | 0.003 | 0.071 | 0.036 |

**Table 2.** Chemical composition (wt. %) of 304 stainless steel.

| Fe | C | Si | Mn | P | S | Ni | Cr | Co |
|---|---|---|---|---|---|---|---|---|
| Bal. | 0.06 | 0.24 | 1.12 | 0.037 | 0.025 | 10.08 | 18.14 | 0.18 |

### 2.2. JHDB

CP-Ti and 304 stainless steel were bonded by JHDB. The cylindrical rods were horizontally placed on the same axis in the vacuum chamber of a JHDB machine (ECO-A, Moriya, Japan) (Figure 1). Joule heating was performed to rapidly heat the interface of the abutting rods at a constant axial pressure during JHDB. A modified version of a commercial resistance spot welding source (IS-110A, AMADA, Isehara City, Japan) with the maximum power capacity of 130 kVA was used in JHDB. The temperature at the interface was measured by an infrared thermometer (JAPANSENSOR, Tokyo, Japan) and regulated with a feedback circuit using the measured temperature data. Thus, the temperature during the entire process was maintained at the desired level without overshooting. However, it was very difficult to repeat the bonding procedures at exactly the same bonding temperature because JHDB was still under the development. Therefore, the relationship between the interfacial microstructure and the tensile strength of the joints were examined using the multiple samples produced at the similar conditions.

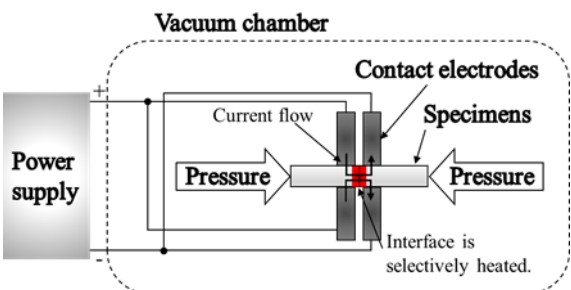

**Figure 1.** Schematic of Joule heating diffusion bonding.

The JHDB of CP-Ti to 304 stainless steel was performed at an axial pressure of 10–50 MPa, a bonding time of 5–25 s, and a bonding temperature of 580–1360 °C. The temperature range used in the present study was selected based on the previous studies on the diffusion bonding of Ti (or its alloys) to steels. The sets of bonding parameters that were used in this study are listed in Table A1.

### 2.3. Mechanical and Microstructual Tests

The tensile strengths of the as-bonded joints without any post-machining, i.e., the cylindrical bonded samples, were examined using a screw-driven tensile tester (AGX-V, Shimadzu, Japan) with a crosshead speed of 1 mm/min at room temperature. The specimens for the microstructural examinations were cut from the joint in a direction perpendicular to the joint interface. Subsequently, they were polished to a 1 μm diamond finish and observed by optical microscopy (Nikon, Tokyo, Japan). The bonding of the entire interface was not achieved in some cases in this study. The ratio of the length of the bonded interface to that of the entire interface was defined as the bonding ratio that represented the degree of bonding. The length of the bonded interface was quantitatively measured on the as-polished cross section by optical microscopy (Nikon, Tokyo, Japan). The microstructure

of the bonded interface was examined by scanning electron microscopy (SEM) (JEOL, Akishima, Japan). The chemical composition of the reaction phases was analyzed by energy-dispersive X-ray spectroscopy (EDS) (EDAX, Mahwah, NJ, USA) in conjunction with SEM. The thickness of the reaction phases was quantified at the 20 locations of the interface in a SEM image. A foil of the interface was thinned by focused ion beam (FIB) milling, and the thin foil was observed by transmission electron microscopy (TEM) (JEOL, Akishima, Japan) at an accelerating voltage of 200 kV. The fracture surfaces of the joints were subjected to X-ray diffraction (XRD) to determine the location of the fracture during tensile testing.

## 3. Results and Discussion

### 3.1. Appearnace of the Joint

Figure 2 shows the appearance of the joint that was bonded at an axial pressure of 30 MPa, a bonding time of 15 s, and a bonding temperature of 1140 °C. The joint exhibited a high tensile strength. The appearance indicated that there was approximately no macroscopic deformation of the workpiece, despite the low bonding time. Conversely, the friction-welded joints show large flashes [13]. Therefore, JHDB effectively prevented the macroscopic deformation of the workpieces and lowered the processing time.

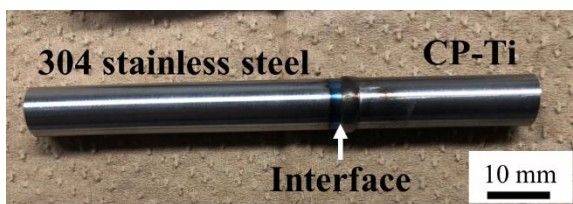

**Figure 2.** Appearance of the 304 stainless steel/CP-Ti JHDBed joint (axial pressure, bonding time, bonding temperature of 30 MPa, 15 s, and 1140 °C, respectively).

### 3.2. Tensile Strength

Figure 3a shows an appearance of the typical joint after the tensile failure. All joints hardly experienced the plastic deformation and failed at the joint interface, as shown in Figure 3a. Macroscopic images of the fracture surfaces of the joints exhibiting the high and low tensile strengths are presented in Figure 3b. The flat fracture surfaces are found in all the joints, but initial patterns produced by machining for the sample preparation are partly left on the fracture surfaces of the joints with the low tensile strength.

Figure 4 shows the effects of the bonding parameters on the tensile strength of the joints. It should be noted that Figure 4 was made with all data obtained from all joints produced at various bonding parameters, i.e., it is not the single effect of each bonding parameter on the tensile strength, so that the results in this figure are largely scattered. Figure 4a,b exhibited scattered tensile strengths, and there was no clear dependence of the axial pressure and the bonding time on the tensile strength. However, as shown in Figure 4c, a relatively good correlation between the tensile strength and the bonding temperature was surprisingly obtained, i.e., the tensile strength virtually increased with the increase in the bonding temperature up to 1100 °C and decreased thereafter. This result implies that the bonding temperature is the major determining factor for the tensile strength of the joints produced by JHDB.

### 3.3. Microstructures

Figure 5 shows the macroscopic images of the interfaces of the joints that were bonded at 880 °C and 1140 °C. A substantial portion of an insufficiently bonded joint interface (shown in Figure 5) existed primarily at the periphery of the joint that was bonded at 880 °C. The bonding ratio for this joint was calculated to be 91%. The bonding ratio for the joint that was bonded at 1140 °C was approximately

100%. A thorough analysis revealed that the bonding ratio increased with the increase in the bonding temperature up to 1100 °C; subsequently, the bonding ratio saturated at approximately 100% beyond 1100 °C (Figure 6).

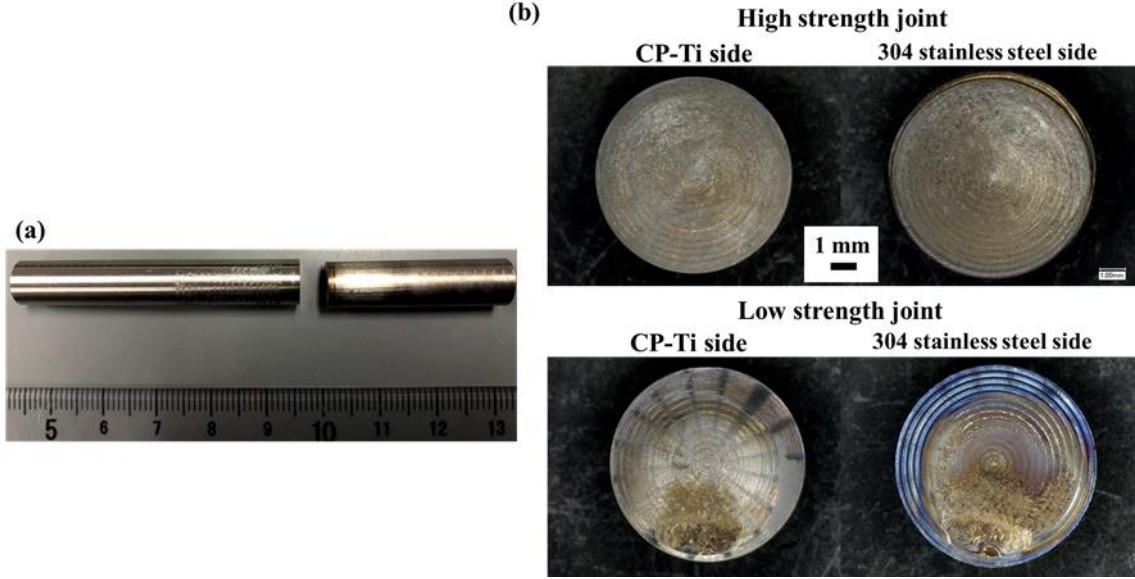

**Figure 3.** (**a**) Appearance and (**b**) fractured surfaces of the joints after tensile testing.

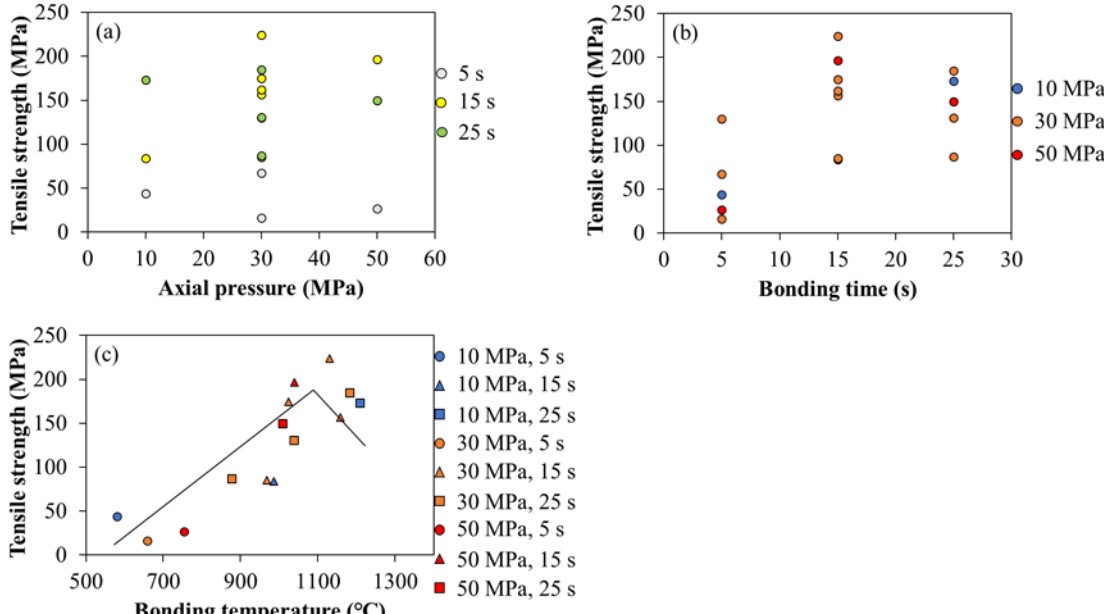

**Figure 4.** Effects of the (**a**) axial pressure, (**b**) bonding time, and (**c**) bonding temperature on the tensile strength of the joint.

The tensile strength increased with the increase in the bonding temperature up to 1100 °C and decreased thereafter (Figure 4). Therefore, the bonding temperature exerted similar effects on the tensile strength and the bonding ratio. Multiple previous studies on the solid-state bonding of dissimilar materials showed an increase in the bonding strength with an increase in the bonding ratio (fraction of the sufficiently bonded interface) [15–18]. These outcomes suggested that the tensile strengths of the joints in this study could also be explained based on the bonding ratios; however, the variations in the tensile strength that were greater than 1100 °C were inconsistent with the tendencies that were indicated in the previous studies.

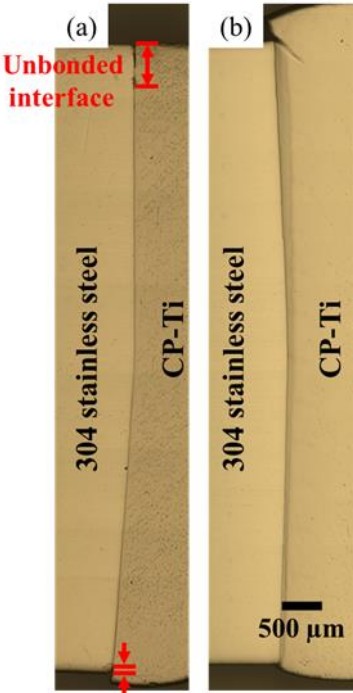

**Figure 5.** Optical micrographs of the 304 stainless steel/CP-Ti joint at an axial pressure of 30 MPa, a bonding time of 15 s, and a bonding temperature of (**a**) 880 °C and (**b**) 1140 °C.

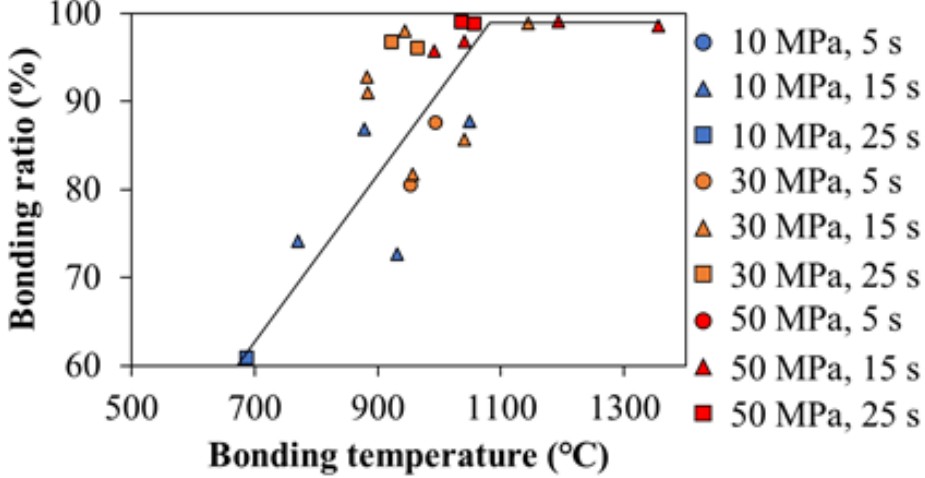

**Figure 6.** Effect of the bonding temperature on the bonding ratio.

The interfacial microstructure was subjected to detailed examinations to elucidate the reasons for the inconsistency at high bonding temperatures.

Figure 7 shows a typical backscattered electron (BSE) image of the joint interface. Three reaction layers, which were labeled as phases 1 to 3, were observed at the joint interface. The layers of phases 1 and 2 resembled the typical reaction layers at a dissimilar interface. The wide layer of phase 3 exhibited a similar contrast to that of CP-Ti. The total thickness of the reaction phases at the joint interface increased substantially with the increase in the bonding temperature owing to the increase in the interdiffusion of the alloying elements [3,19]. The increase in the bonding time induced an insignificant increase in the total thickness of the reaction phases. The axial pressure exerted a negligible effect on the total thickness of the reaction phases, which was consistent with the observations in the previous studies on conventional diffusion bonding [19].

The chemical composition of the reaction phases was examined by SEM/EDS. Phase 1 contained ~62 at.% Fe, ~5 at.% Ti, ~27 at.% Cr, ~4 at.% Ni, and ~2 at.% Mn. Therefore, phase 1 was identified as an Fe-based metallic phase. Phase 2 contained ~37 at.% Fe, ~50 at.% Ti, ~7 at.% Cr, ~5 at.% Ni, and ~1 at.%. Mn. The Fe-Ti phase diagram indicated the presence of IMCs, i.e., FeTi and/or $Fe_2Ti$, in phase 2 [4]. Phase 3 contained ~10 at.% Fe, ~86 at.% Ti, ~2 at.% Cr, ~2 at.% Ni, and < 1 at.% Mn. Therefore, phase 3 was identified as β-Ti because Fe, Cr, and Ni, which are the major alloying elements in 304 stainless steel, were β-stabilizers of Ti.

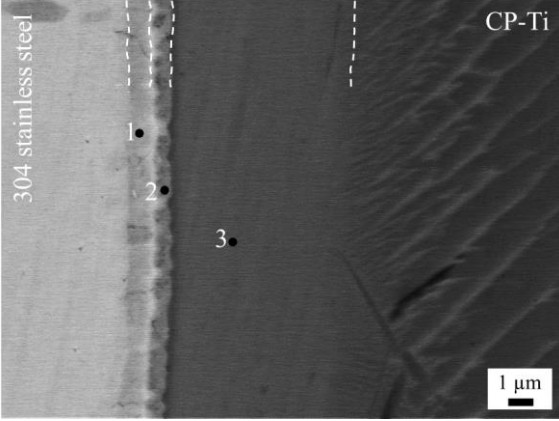

**Figure 7.** A typical scanning electron microscopy backscattered electron (SEM-BSE) image of the JHDBed 304 stainless steel/CP-Ti joint (axial pressure, bonding time, and bonding temperature of 10 MPa, 25 s, and 1240 °C, respectively).

The bright field (BF) image of the joint interface (Figure 8a) showed four layers with a total thickness of 1.5 μm that possibly corresponded to the layers of phases 1 and 2 (Figure 7). The selected area electron diffraction (SAED) patterns that were obtained from layers A–D are shown in Figure 8b–e, respectively. The chemical compositions of the layers in Figure 7 and the SAED identified layers A–D in Figure 8a, such as α-Fe, $Fe_{16}Cr_8Ti_5$, $Fe_2Ti$, and FeTi, respectively.

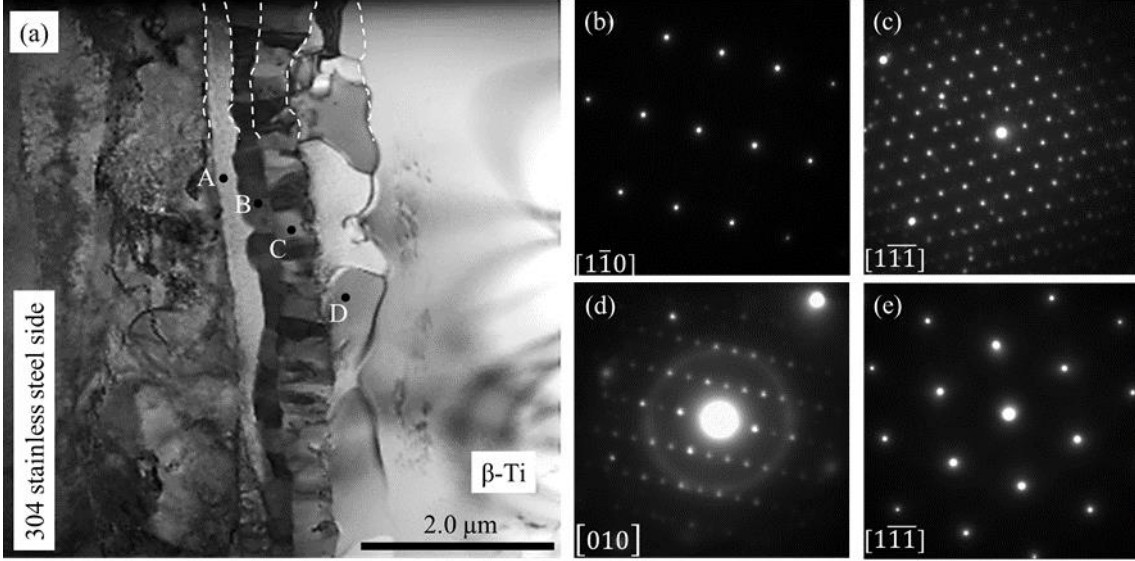

**Figure 8.** (**a**) Transmission electron microscopy bright field (TEM-BF) image of the joint interface. Selected area electron diffraction (SAED) patterns of layers (**b**) A, (**c**) B, (**d**) C, and (**e**) D.

### 3.4. XRD Analysis of the Fracture Surface

Figures 9 and 10 show the results of the XRD analysis for the joints that were produced at 970 °C and 1160 °C, respectively. The XRD spectra were obtained from the fracture surfaces of the 304 stainless-steel side and the CP-Ti side of the joints. $\gamma$-Fe was detected on the 304 stainless-steel side, while $\alpha$-Ti and $\beta$-Ti were detected on the CP-Ti side of the joint that was bonded at 970 °C. This was attributed to the insufficient bonding between 304 stainless steel and CP-Ti (Figure 6). The XRD spectra for the joint that was bonded at 1160 °C showed peaks corresponding to $Fe_{16}Cr_8Ti_5$, $Fe_2Ti$, FeTi, and $\gamma$-Fe on the 304 stainless-steel side; additionally, peaks corresponding to $Fe_2Ti$, FeTi, $\alpha$-Ti, and $\beta$-Ti were observed on the CP-Ti side. It was concluded that the joint failed along the Fe-Ti-type IMC layers (layers C and D in Figure 8) because the peaks of $Fe_2Ti$ and FeTi were detected on the fracture surfaces of both sides.

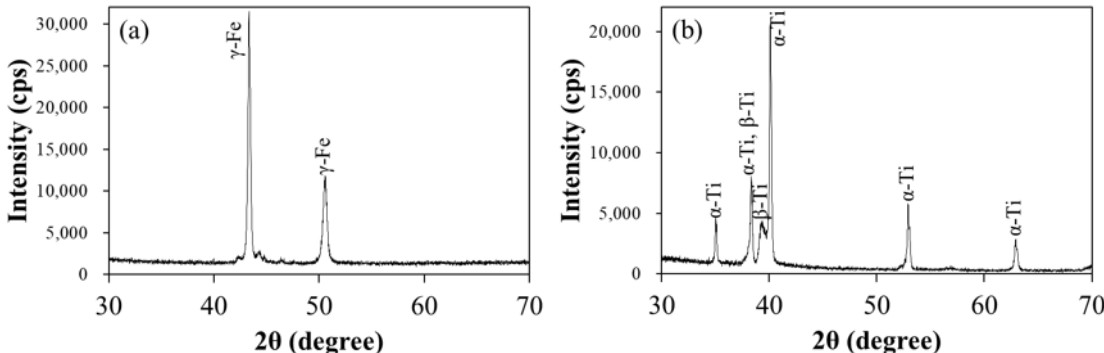

**Figure 9.** XRD spectra obtained from the fracture surfaces of the (**a**) 304 stainless-steel side and the (**b**) CP-Ti side of the joint that was bonded at 970 °C, with an axial pressure and a bonding time of 30 MPa and 15 s, respectively.

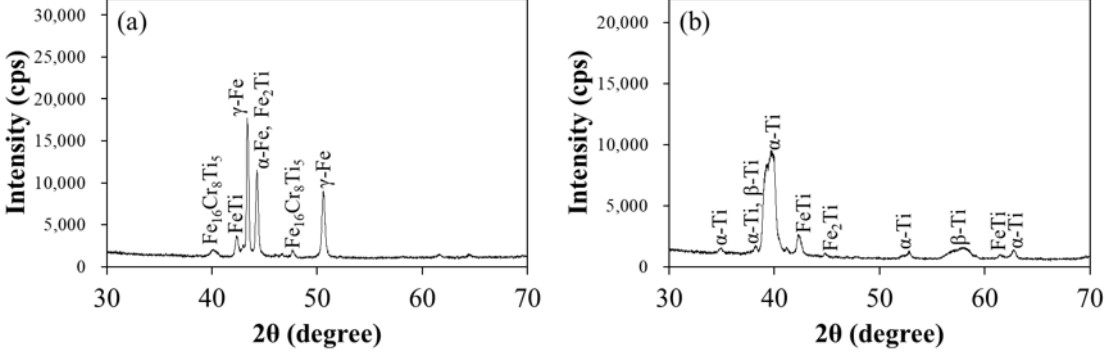

**Figure 10.** XRD spectra obtained from the fracture surfaces of the (**a**) 304 stainless-steel side and the (**b**) CP-Ti side of the joint that was bonded at 1160 °C, with an axial pressure and a bonding time of 30 MPa and 15 s, respectively.

### 3.5. Relationship between the Tensile Strength of the Joint and the Thickness of the Reaction Layer

When the bonding temperature was higher than 1100 °C, the bonding ratio was approximately 100% (Figure 6). The joint with a bonding ratio of approximately 100% exhibited a low tensile strength at high bonding temperatures and failed along the Fe-Ti-type IMC layers during tensile testing. Multiple previous studies reported the failure of the dissimilar joints of Ti and steel along the Fe-Ti-type IMC layers; furthermore, the tensile strength of such joints decreased with the increase in the thickness of the Fe-Ti-type IMC layers [20]. Hinotani examined the effect of the thickness of the Fe-Ti-type IMC layers on the tensile strength of the dissimilar joints of Ti and steel. The results showed the decrease in the tensile strength with the increase in the thickness of the Fe-Ti-type IMC layers to greater than approximately 1 µm [21]. This suggested that the decrease in the tensile strength at high bonding

temperatures depended on the thickness of the Fe-Ti-type IMC layers. Figure 11 shows the effect of the bonding temperature on the tensile strength of the joint and the thickness of the Fe-Ti-type IMC layers. The tensile strength of the joint in this study decreased with the increase in the bonding temperature to greater than 1100 °C, where the thickness of the Fe-Ti-type IMC layers was 0.5 μm. This thickness was close to that reported by Hinotani for the decrease in the tensile strength. Therefore, the decrease in the tensile strength of the joint at high bonding temperatures was attributed to the growth of brittle Fe-Ti-type IMCs at the joint interface. From these results, the present study suggests that the well-bonded interface with the Fe-Ti-type IMC layers which was thinner than 0.5 μm needs to be created to achieve the high tensile strength of the CP-Ti/304 stainless steel joint produced by JHDB.

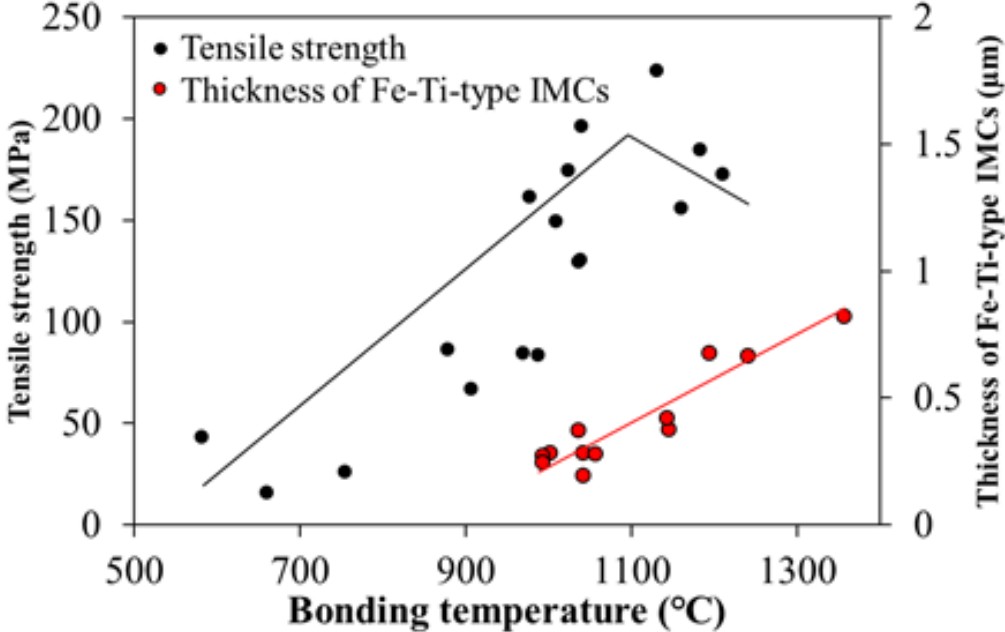

**Figure 11.** Effects of the bonding temperature on the tensile strength of the joint and the thickness of Fe-Ti-type intermetallic compound (IMC) layers.

## 4. Conclusions

This study demonstrated the application of JHDB for the dissimilar bonding of CP-Ti to 304 stainless steel and examined the effect of microstructural factors at the interface on the tensile strengths of the joints. JHDB effectively prevented the macroscopic deformation of the workpieces and lowered the bonding time. The tensile strength of the joints increased with the increase in the bonding ratio. In the joints with a well-bonded interface, the tensile strength of the joints decreased with the increase in the thickness of the $Fe_2Ti$ and FeTi layers. Consequently, the present study suggested that both a bonding ratio higher than 98% and Fe-Ti-type IMC layers thinner than 0.5 μm are required for the high strength in the CP-Ti/304 stainless steel joint produced by JHDB.

**Author Contributions:** Conceptualization, Y.S.S.; methodology, K.T.S., Y.S.S., and S.T.; formal analysis, K.T.S.; investigation, K.T.S.; resources, Y.S.S. and S.T.; data curation, K.T.S., Y.S.S., and S.T.; writing—original draft preparation, K.T.S.; writing—review and editing, Y.S.S. and S.T.; visualization, K.T.S.; supervision, Y.S.S.; project administration, Y.S.S.; funding acquisition, Y.S.S. All authors have read and agreed to the published version of the manuscript.

**Funding:** This research received no external funding.

**Acknowledgments:** The authors are grateful to Kosei Kobayashi and Shun Omura at Tohoku University for their technical assistance, to ECO-A for sample preparation.

**Conflicts of Interest:** The authors declare no conflict of interest.

## Appendix A

**Table A1.** Bonding parameters used in this study.

| Axial Pressure (MPa) | Bonding Time (s) | Bonding Temperature (°C) | Subjected Testing |
|---|---|---|---|
| 10 | 5 | 580 | Tensile test |
| | | 640 | Observation |
| | | 650 | Observation |
| | 15 | 950 | Observation |
| | | 990 | Tensile test |
| | 25 | 1140 | Observation |
| | | 1210 | Tensile test |
| | | 1240 | Observation |
| 30 | 5 | 630 | Observation |
| | | 660 | Tensile test |
| | | 770 | Observation |
| | | 880 | Observation |
| | | 910 | Observation |
| | | 930 | Observation |
| | | 1040 | Observation |
| | | 1050 | Observation |
| | 15 | 880 | Observation |
| | | 940 | Observation |
| | | 960 | Observation |
| | | 970 | Observation |
| | | 980 | Observation |
| | | 1000 | Observation |
| | | 1020 | Tensile test |
| | | 1040 | Observation |
| | | 1130 | Observation |
| | | 1140 | Observation |
| | | 1160 | Observation |
| | 25 | 880 | Observation |
| | | 990 | Observation |
| | | 1040 | Observation |
| | | 1180 | Tensile test |
| | | 1190 | Observation |
| | | 1360 | Observation |
| 50 | 5 | 690 | Observation |
| | | 750 | Tensile test |
| | | 910 | Observation |
| | 15 | 920 | Observation |
| | | 960 | Observation |
| | | 1040 | Tensile test |
| | 25 | 1010 | Tensile test |
| | | 1040 | Observation |
| | | 1060 | Observation |

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
