# Peer review of "Rapid Joining of Commercial-Purity Ti to 304 Stainless Steel Using Joule Heating Diffusion Bonding: Interfacial Microstructure and Strength of the Dissimilar Joint"

_metals, doi:10.3390/met10121689_

Round 1
Reviewer 1 Report
Recently, there have been ample publications on diffusion bonding of Titanium to Stainless Steels, which is the topic this manuscript by Suzuki et al. too. Although the details of the bonding procedure are not given, I assume the authors used a Thermo-Mechanical Simulator (e.g. Gleeble) to bond the samples within a very short time (a few seconds). I am unable to recommend this manuscript for publication due to the following shortfalls.
1- The manuscript does not refer to recent developments in the field, particularly the new variants of diffusion bonding which produced far stronger joints (+90% of Ti) than those achieved in this work. Please Google “diffusion bonding of stainless steel to titanium” to find out more.
2- The manuscript fails to provide a fair and critical assessment of the process used in the work. Therefore, the comments made on the superiority of the new method in comparison with conventional diffusion bonding are incorrect. For instance, the process relies on passing very high electrical currents and applying high pressures i.e. 50 to 250 kg load on a thin 8mm diameter rods. Such extreme bonding conditions can only be generated in simulators and on small samples. Yet, the work shows achieving full bonded areas was a challenge even on small samples. The application on the proposed process is mostly for research purpose due to such limitations.
Subject to ratifying above issues (1&2) in the abstract, text and conclusions, and providing more detailed experimental procedure (e.g. the bonding rig and current density, voltage) the article might be considered for publications (Mandatory).
XRD charts are not needed and they make reading the lengthy article tiresome (optional).
By the way, Joule Heating is a terminology mostly used by Physicists. The common engineering terminology is Electrical Resistance Heating, which is the same as in any ordinary hair dryer or electric heater (optional).
Reviewer 2 Report
In general, the subject of the paper is very interesting and it is a very promising area of research, especially, when the scaleability issue will be resolved. At the same time, there are significant gaps in the experimental validation of the results and conclusions, provided in the current state of the paper:
p1 l.17 - the author said the JHDB method is novel - although there are mentioning of similar technologies in the publications from 2012. Potentially author would like to point to the novelty of the application? or combination of the materials, but it wasn't reflected at all.
p1 l.34 - the author said: Ti is expensive.. compared to what? Other materials like tailored Al alloys and Ni alloys could be even more expensive. Also, cost reduction could be not the main driver as other counterpart alloys might significantly affect the properties of the whole joint.
p1 l.41 - authors need to mention what does the "large" means? it will help readers to get the right dimensional feeling
p2 l.48 - 3MPa - digit and unit should be in one line
p2. l87-88 - Table and 1 should be in one line and should be a space between Table and 2
p 3-4 Table is very long, can it be removed to the appendix?
p4 l.112 - "universal tensile tester" is not descriptive enough? how does it look like? what was the same geometry? why authors didn't refer to any of the standards that usually used for tensile tests?
P4 l125 - last line on the page should be moved to the next page
p5 l 128 - what was the basis for this conclusion? no evidence? no supportive analysis so far?
p5 l139 - it will be good to show some images of the failed joints
p5 l143 - add reference to fig 3c.
Fig 3 - the presented data is very scattered and strength measurements are not really representative. Also, it is very hard to distinguish different conditions and graph should be provided in colour. If the data so scattered does it mean that the welding method is not consistent or the testing methodology is not accurate enough?
Figure 4a - it is not clear from the picture that areas are well bonded and it is hard to confirm from the provided image that bonding ration is 91%.
Figure 4b - there is the crack on the top of the image, that evidence about deformation induced by excessive load for selected temperature/time conditions
p6 l158 - there is a tendency for increased tensile strength but scatter of the data is too high to conclude about increased strength
p6 l160 - reference needed
p6 l162-165 - there is in general a very big question about the testing methodology? how consistent it is?
p6-7 l182-185 - where is the data that supports this statement?
Author compares joint micrographs for the conditions bonded at 880C and 1140C and have a fracture analysis for conditions bonded at 970C 1160C. From this point of view the overall analysis looks very inconsistent and provides very low statistics.
It is advised that authors would better repeat 2-3 times bonding tests for similar or best identified conditions and provide full analysis i.e. microstructure mechanical properties and X-ray for these joints.
The conclusion in the section 3.5 based only on correlation of the temperature and tensile strength, meanwhile a very broad scatter of the tensile data also might be influenced by the applied load, which was not considered.
If author would like to compare the temperature vs UTS only, than it should be done for the same load.
Also, it wasn't particularly described how the thickness of the bonded layer and the bond length have been measured.
As a final suggestion, the paper might significantly benefit if fractography analysis will be provided and fracture mechanics will be discussed.
Reviewer 3 Report
- Abstract does not describe the content of the article, it provides general theoretical information. We recommend shortening it and providing expert information on the issues addressed in the article.
- The introduction needs to be reworked, it is written very generally. It is not possible to quote every word in the section: Titanium (Ti) and its alloys have several desirable properties, such as high specific strength [1], high corrosion resistance [2], low density [3] and low thermal conductivity [ 4]. Therefore, they are suitable for various applications in the chemical [5], aerospace [6], nuclear [7], medical [8], military [9], cryogenic [10], petrochemical [11] and energy [12] industries. . Because it is dear to you, its usefulness is often limited [13]. This requires the combination of Ti with cost-effective materials such as stainless steel [14]. ......This is not a serious literary search and citation.
- It is necessary to specify the basic material 304 stainless steel, properties and area of application, ISO standard.
- Table 3 - it is necessary to specify why these temperatures were chosen and why such a large temperature range was used
- Fig.3 - Bondong tine (s) - repair
- Fig.3 - Graphs are not clearly processed, the scatter of measured data is missing. The result of the measurements is little described in the text.
- Very brief conclusion, does not capture experimental research results. The conclusion needs to be reworked and the results supplemented. There is a lack of confrontation with modern magazines and patent literature in international sources.
Reviewer 4 Report
A submission on diffusion bonding of cp Ti and 304 ss. The following comments can be made:
- the experimental setup and the power source used need further details
- in fig.3 the points can hardly be seen
Round 2
Reviewer 1 Report
The revised manuscript merits publication in this journal.
Reviewer 2 Report
The authors carefully reviewed the provided comments and suggestions and significantly improved the paper. I would strongly recommend keep working on the bonding methodology, as the repeatability of the bonding parameters is a key factor.